# Recommendations for an exercise intervention and core outcome set for older patients after hospital discharge: Results of an international Delphi study

Jesse J. Aarden[1,2,3]*, Mel E. Major[1,2,3], Claartje M. W. Aghina[2], Martin van der Esch[2,4], Bianca M. Buurman[2,5], Raoul H. H. Engelbert[1,2], Marike van der Schaaf[1,2]

1 Department of Rehabilitation, Amsterdam Movement Sciences, Amsterdam UMC, University of Amsterdam, Amsterdam, The Netherlands, 2 Centre of Expertise Urban Vitality, Faculty of Health, Amsterdam University of Applied Sciences, Amsterdam, The Netherlands, 3 ESP—European School of Physiotherapy, Faculty of Health, Amsterdam University of Applied Sciences, Amsterdam, The Netherlands, 4 Reade, Center for Rehabilitation and Rheumatology/Amsterdam Rehabilitation Research Center, Amsterdam, The Netherlands, 5 Department of Internal Medicine, Section of Geriatric Medicine, Amsterdam Public Health Research Institute, Amsterdam UMC, University of Amsterdam, Amsterdam, The Netherlands

* j.j.aarden@hva.nl

## Abstract

For older adults, acute hospitalization is a high-risk event with poor health outcomes, including functional decline. In absence of practical guidelines and high quality randomized controlled trials, this Delphi study was conducted. The aim of this study was to obtain consensus on an exercise intervention program, a core outcome set (COS) and handover information to prevent functional decline or restore physical function in acutely hospitalized older patients transitioning from hospital to home. An internal panel of experts in the field of exercise interventions for acutely hospitalized older adults were invited to join the study. In the Delphi study, relevant topics were recognized, statements were formulated and ranked on a 9-point Likert scale in two additional rounds. To reaching consensus, a score of 7–9 was classified as essential. Results were expressed as median and semi-interquartile range (SIQR), and consensus threshold was set at SIQR≤0.5. Fifteen international experts from eight countries participated in the panel. The response rate was 93%, 93% and 80% for the three rounds respectively. After three rounds, consensus was reached on 167 of the 185 (90.3%) statements, of which ninety-five (51.4%) were ranked as essential (median Likert-score ≥7.0, SIQR ≤0.5). This Delphi study provides starting points for developing an exercise intervention, a COS and handover information. The results of this Delphi study can assist physical therapists to provide a tailored exercise intervention for older patients with complex care needs after hospital discharge, to prevent functional decline and/or restore physical function.

**Data Availability Statement:** All relevant data are within the paper and its Supporting Information Files.

**Funding:** YES: This research is supported by a doctoral grant for J.J. Aarden from NWO, the Netherlands organization for scientific research (No 023.011.059).

**Competing interests:** NO: the authors have declared that no competing interests exist.

## Introduction

For older adults, an acute hospitalization for multiple days due to an acute illness is a high-risk event with poor health outcomes, including functional decline, readmission, and mortality [1]. More than 30% of older adults experience physical deconditioning and functional decline after acute hospitalization [2, 3]. Several factors are associated with functional decline, including severity of the acute disease, immobility [4, 5], reduced physical activity [6, 7], low muscle mass/strength [8, 9], nutritional deficiency [10] and geriatric syndromes [11, 12]. These factors are highly prevalent in older patients after acute hospitalization and might hinder recovery, reduce physical functioning and promote functional decline [3, 11].

Functional decline is the loss of activities of daily living with worsening self-care skills [13] and can be reduced during hospitalization with an exercise program [14]. In this study [14], in-hospital exercise programmes to prevent functional decline were performed twice per day. These programmes included multiple components that focused on the patients' individual needs [14]. Providing older patients with an exercise programme when they transition from the hospital to home has been associated with better recovery and less functional decline. However, this association has not been confirmed [15–17]. Exercise interventions started in the hospital are often not continued at home, despite the importance of these interventions to the patients [2].

A seamless transition of exercise interventions from the hospital to home might stimulate recovery and prevent functional decline [1, 11, 13]. International exercise recommendations in older adults are reported [18, 19], These recommendations indicate that exercise improve physical function and quality of life and exercise is essential to older adults. However, practical guidelines on the frequency, intensity, time, and type (FITT) of home-based exercise interventions specifically for older patients after hospitalization are lacking. Also important to a seamless transition in rehabilitation care from hospital to home are recommendations for handover information and measurement tools as part of a core outcome set (COS) for clinical practice. It has been suggested that a COS would increase uniformity [19–22] in research and clinical practice and might help create exercise intervention programmes that are tailored to the individual needs and goals of the patient.

In the absence of practical guidelines and high-quality randomized controlled trials focusing on acute hospitalized older adults, the Delphi methodology is often applied to obtain expert consensus on interventions for different populations [23, 24]. If experts could agree on practical guidelines for an exercise intervention, a COS and handover information for older patients after acute hospitalization in the home situation, this would guide physical therapists in their clinical decision-making. The aim of this Delphi study was to develop a consensus statement on 1) the characteristics of a home-based exercise intervention, 2) a COS of measurements on daily functioning and 3) handover information for older, acutely hospitalized patients transitioning from hospital to home that can prevent functional decline or restore physical function.

## Methods

To determine topics relevant to the objective of this Delphi study, a scoping literature review was conducted on measurement tools and exercise interventions for older adults. After this, a three round Delphi method was applied. A steering committee consisting of experts in complex care and rehabilitation after acute hospitalization from the Amsterdam University Medical Centers (Amsterdam UMC) supervised the Delphi project. The project was registered with the Core Outcome Measures in Effectiveness Trials (COMET) initiative (study reference: http://www.comet-initiative.org/Studies/Details/1294).

| Unable to score | Limited importance | | | Importance, but not essential | | | Essential | | |
|---|---|---|---|---|---|---|---|---|---|
| O | 1 | 2 | 3 | 4 | 5 | 6 | 7 | 8 | 9 |

**Fig 1. 9-point Likert scale used in the Delphi rounds.**

We conducted a scoping literature review searching PubMed, Medline, PEDro, CINAHL, Science Direct and ProQuest Social Sciences to summarize the current state of the art [24, 25]. This scoping review included studies on characteristics of exercise interventions and measurement tools within the domains of the International Classification of Functioning (ICF) [26] for older patients after acute hospitalization. Articles were considered for review if they were systematic reviews or clinical trials and published in the last 10 years and if exercise for older adults was the studied intervention. Based on the scoping review, the following three topics were recognized: 1) characteristics of the exercise intervention, 2) COS of measurement tools and 3) handover information from hospital on to healthcare professionals in primary care. Statements on the three topics were formulated and then discussed by the panel.

## Expert panel

Delphi panel members were recruited based on their clinical and scientific expertise in exercise interventions, their professional background, their research output, and their geographical location. Eligible panellists were invited to participate via email, and informed consent for publication of the results was obtained when they agreed to participate.

## Delphi rounds

The Delphi rounds were conducted between January and April 2019. It was decided, a priori, to conduct a minimum of three rounds because this is considered appropriate when limited scientific evidence is available [24]. A digital survey was sent to generate ideas and to rank statements on a 9-point Likert scale, as per Delphi methodology recommendations [20]. A score of 1–3 was given to items of limited importance; a score of 4–6 to items ranked as important but not essential; and a score of 7–9 to items deemed essential (Fig 1). Panellists could also give a score of 0 (unable to score) if they felt a topic or statement fell outside of their scope of expertise. For each statement scored in the second and third Delphi rounds, a median Likert-score and semi-interquartile range (SIQR) were computed based on the first and third quarters of the SIQR. Results from the second Delphi round were imputed into the final round results if no third-round score was given. Consensus was defined a priori as an SIQR $\leq 0.5$. Statements with consensus and a median Likert score $\geq 7.0$ were used for further analysis. Consensus was reached on $\geq 80\%$ of the statements after round three, so no extra Delphi round was deemed necessary [23–25].

## Delphi round 1: Collecting expert opinions

The aim of the first round was to collect expert opinions on the three topics identified in the scoping review (exercise intervention, COS, and handover information). A case description of an acutely hospitalized older adult transitioning home from hospital provided the context and was the starting point for each panel member (supplementary material). The questions were related to the different aspects of the ICF and used a standard description of health and health-related status [26]. In this first round, 22 closed questions on the three topics were asked with multiple possible answers. Additional information was also collected from 17 open questions on topics such as the intensity of training or involvement of other healthcare professionals

(supplementary material). All items checked as relevant by the panel members were included in the following rounds. Answers to the open questions were examined to check whether they raised new questions or identified different topics. All input was categorized, and statements were drafted for each of the topics and approved by the steering committee.

### Delphi round 2: Ranking statements

After the first Delphi round, 185 statements were formulated: 74 on exercise interventions, 86 on measurement tools and COS, and 25 on handover information.

### Delphi round 3: Consensus round

In the third Delphi round, each panellist received their results from the second Delphi round together with the panel's median Likert scores and SIQR for each of the statements. If an individual panel member's scores differed from the panel's median scores, they were asked to consider re-ranking the statement towards the median to reach consensus. Participants were motivated further if they chose not to re-rank their statements.

## Results

All invited experts agreed to participate in the Delphi panel (n = 16). One panellist did not respond within the allocated time for the first Delphi round so 15 panellists were included in the analysis. The response rates were 93% for round one, 93% for round two and 80% for round three. Table 1 presents the panellists' nationalities, profession, field of expertise, years of clinical experience and response. The panel consisted of nine physical therapists, two exercise physiologists, two sports scientists, one physician and one occupational therapist. After round three, consensus was reached on 185 statements, warranting the end of the Delphi consensus process. Ninety-five of the 185 statements (51.4%) were consensually ranked between 7 and 9 on the Likert scale and therefore considered essential for implication in clinical practice by the Delphi panel.

### Theme 1: Exercise intervention

Seventy-four of the 185 (40.0%) statements were about exercise interventions to prevent functional decline after hospital discharge. Of these, 55 statements (74.3%) were consensually ranked as essential (supplementary material). Statements covered topics such as FITT of training, the need for supervised exercise programmes, importance of exercise programmes, and whether exercise interventions should be combined with nutritional and behavioural interventions. Regarding training frequency, daily exercise interventions in the acute phase (up to 7 days post-discharge), 2–3 times weekly interventions in the sub-acute phase (up to 12 weeks post-discharge) and 1–12 times weekly in the long-term phase (>12 weeks post-discharge) were consensually ranked as essential for preventing functional decline. The panel agreed that exercise intensity levels up to 70–80% of the maximum heart rate are essential for preventing functional decline and that contra-indications should be absent. With regards to the type of training in the acute phase, the panel ranked early mobilization, supervised tailor-made exercise interventions adjusted to the specific needs and goals of the patient, and combined exercise interventions (including strength, aerobic and functional training, either individual or in a group) as essential. Furthermore, co-creation of a training program by the patient and healthcare professional, functional training, building up physiological reserves, coaching, and reassessment and treatment by a geriatrician post-discharge were all ranked as essential during the recovery phases. Fig 2 summarizes these exercise intervention characteristics and existing recommendations.

**Table 1. International Delphi panel characteristics.**

| | Country | Gender | Title | Field of Expertise | Years of Clinical experience | Number of Publications in PubMed | Round 1 | Round 2 | Round 3 |
|---|---|---|---|---|---|---|---|---|---|
| 1 | Australia | Male | Professor | Exercise physiologist | >20 | >100 | ✓ | ✓ | ✓ |
| 2 | Belgium | Male | MSc | Physical therapist | >20 | >5 | ✓ | ✓ | ✓ |
| 3 | Belgium | Male | Professor | Exercise Physiologist | 10–15 | >100 | ✓ | ✓ | ✓ |
| 4 | Canada | Female | PhD | Physical therapist | 5–10 | >50 | ✓ | ✓ | ✓ |
| 5 | Denmark | Female | PhD | Physical therapist | 15–20 | 10 | ✓ | – | – |
| 6 | Netherlands | Female | MSc | Physical therapist | 15–20 | 0 | ✓ | ✓ | ✓ |
| 7 | Netherlands | Female | MSc | Physical therapist | 15–20 | 0 | ✓ | ✓ | ✓ |
| 8 | Netherlands | Female | PhD | Physical therapist | 10–15 | >50 | ✓ | ✓ | ✓ |
| 9 | Netherlands | Male | PhD | Exercise physiologist | 10–15 | >50 | ✓ | ✓ | ✓ |
| 10 | Netherlands | Female | MSc | Physical therapist | >20 | 0 | ✓ | ✓ | ✓ |
| 11 | Spain | Female | Associate professor | Physical therapist Sport scientist | 10–15 | >50 | ✓ | ✓ | ✓ |
| 12 | Spain | Male | Professor | Sport scientist | 15–20 | >100 | ✓ | ✓ | – |
| 13 | Spain | Male | Associate professor | Medical doctor | >20 | >100 | ✓ | ✓ | – |
| 14 | USA | Female | Associate professor | Occupational therapist | 10–15 | >20 | ✓ | ✓ | ✓ |
| 15 | USA | Male | Associate professor | Physical therapist | 5–10 | >10 | – | ✓ | ✓ |

– = no response; ✓ = response obtained

### Theme 2: Core outcome set

Eighty-six of the 185 (46.5%) statements were related to measurement tools for the COS. Of these statements, 25.6% (22 statements) were consensually ranked as essential. For activities of daily living, functional exercise capacity, performance, and muscle strength, more than one measurement outcome was ranked as essential. Fig 3 presents an overview of the measurement tools across all ICF domains. A COS of measurement tools was consensually ranked as essential for identifying risk factors of functional decline.

### Theme 3: Handover information

Of the 185 statements, 25 (13.5%) were related to the handover information provided when the patient is discharged from hospital. The panel consensually ranked five demographic aspects as essential for inclusion in handover information: age, gender, weight, height and living situation. Panellists also ranked the following 13 items as essential for inclusion in the handover information: hospital length of stay, number of days of bedrest and sedentary behaviour, comorbidities, reason for hospital admission and/or severity of illness, medication usage, physical therapy interventions, level of (physical) functioning at hospital discharge, premorbid level of functioning, nutritional intake, and treatment goals. Detailed ranking results including median Likert scores and SIQRs can be found in the supplementary material.

### Discussion

This Delphi study provides practice guidelines for an exercise intervention, a COS and handover information to facilitate the seamless transition of exercise interventions when older patients are discharged from hospital. Experts agreed that supervised intensive exercise programmes should continue after hospital discharge and that these interventions should be

**Discharge information**

*Demograhic:*
- Age, gender,
- living situation
- Weight, height

*Other:*
- Reason admission
- Severity of disease
- Comorbidities
- Medication
- Days in hospital
- Sedentary behavior
- Nutritional intake
- Intervention plan

**Specific recommendations for older patients post-discharge**

- Combined exercise intervention (strength, aerobic, functional) with nutritional & behavioral aspects
- Supervised combined exercise intervention for individual/group in practice/home situation
- Progressive training up to 70-80% maximum heart rate in absence of contra-indications
- Work multi- or interdisciplinary with physical therapist, geriatrician and/or dietician

| | | |
|---|---|---|
| • Early mobilization<br>• Tailored-made<br>• Daily exercises | • Functional independence<br>• Building up reserves<br>• Exercises 2 - 3 times / week | • Coaching<br>• Home-based program<br>• Exercises 1 - 2 times / week |
| **Acute phase**<br>(1-7 days post-discharge) | **Subacute phase**<br>(7 days - 12 weeks post-discharge) | **Long term phase**<br>(>12 weeks post-discharge) |

**Identification potential barriers of recovery**

*Physical factors:*
- Muscle mass
- Muscle strength
- Nutrition

*Psychological factors:*
- Depressive symptoms
- Fatigue
- Fear of falling

| **Patient centered** | **Co-creation patient-professional** | **E-Health** |
|---|---|---|
| • Patient perspective<br>• Tailored-made | • Goal setting with patient / spouse<br>• Shared decision making | • Home-based |

**Existing general recommendations for all phases (ACSM guidelines)**

| **Frequency** | **Intensity** | **Type** | **Time** |
|---|---|---|---|
| • ≥ 3 - 5 days/week with moderate to high intensity | • 5 - 8 on 10 point-scale<br>• ↑ Muscle strength: 8 - 10 exercises, ≥ 1 set of 10 - 15 reps | • Aerobic training<br>• Multicomponent exercises for major muscle groups | • 100 -150 min / week |

**Hospital discharge ⟶ Home**

**Fig 2. Recommended exercise intervention characteristics and handover information derived from this Delphi consensus process in addition to general recommendations for older patients after discharge from hospital.**

tailored to the specific needs of the patient. COS measurement tools in all domains of the ICF and handover information from the hospital can help to tailor the exercise intervention to promote recovery, prevent functional decline, and restore physical function.

After discharge from hospital, exercises and physical activity are often not continued because stimulus by staff or community [22] and/or self-discipline [17] are lacking. The expert panel agreed that an exercise intervention with FITT criteria should be continued after discharge to prevent functional decline or restore physical function. This is consistent with the guidelines on exercise from the American College of Sports Medicine [27, 28] and other exercise recommendations [18, 19]. Exercise interventions are associated with higher activities of daily living [29], better mental health [30] and improved quality of life in older adults. Our panellists also agreed that high-intensity exercise interventions are suitable in this population if no contra-indications are present such as decompensated congestive heart failure or severe aortic stenosis [31]. Therefore, high intensity exercises can certainly be considered to restore physical function in line with the international exercise recommendation [18] Exercise interventions to regain physical functioning should be supervised by a physical therapist in older patients who are discharged from hospital with multiple chronic diseases. This is in line with the recommendation from Echeverria et al. [17]. that home-based programmes require self-discipline, and that group exercise may have an important social element. A novel finding of our study is the expert consensus that tailored exercise interventions should be tuned to the specific needs and goals (such as independent self-care, cooking or gardening) of the patient. Previous research has also suggested setting collaborative goals for complex care interventions in older patients with chronic diseases or multimorbidities [32, 33].

A COS in all domains of the ICF can give a complete overview of an older patient's physical functioning when they return home. Geriatric syndromes such as apathy, fear of falling, fatigue, depressive symptoms [11] or undernutrition [10] are highly prevalent in older patients and prevent recovery of functioning after acute hospitalization [11]. Indicating that these syndromes are present in the handover information when a patient is discharged home from

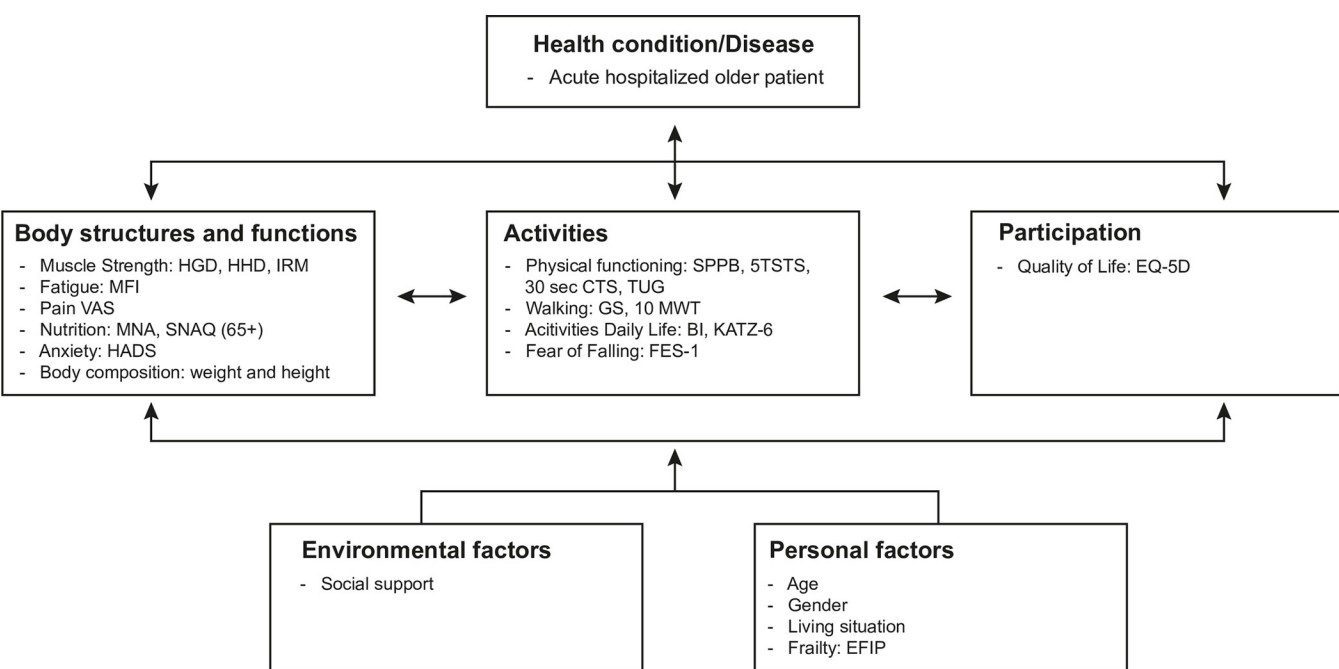

**Fig 3. Core outcome set (COS) of measurement tools per ICF domain post-discharge.**

hospital might increase the success of an exercise intervention. Our expert panel agreed that if multiple risk factors are identified, other healthcare professionals should be involved in the interventions. However, it can be difficult to collect information on all ICF domains of patient functioning because this is time-consuming and burdensome for older patients. Future studies could investigate how to collect this information using wearables [34, 35].

To optimize transitional care, a seamless transition with handover information is important. However, this does not automatically prevent functional decline in older patients [36, 37]. It has been shown that exercise interventions during hospitalization can prevent functional decline or restore physical function [14, 15, 38], but the effects of exercise interventions at home after discharge have not been properly defined [16]. In older patients, the cardiopulmonary and musculoskeletal systems are often not appropriately challenged or loaded by exercise interventions. Finding the optimal FITT training parameters is crucial for recovery [39]. Future research should investigate the effectiveness and appropriateness of exercise interventions and determine how to tailor these interventions to the patient's goals. Our expert panel agreed that eHealth should be investigated in future studies to see whether it can improve the post-discharge care of older patients with complex care needs. Evidence-based knowledge of how psychometric sound assessment tools with normative sex-related values and proper clinical reasoning can be used to tailor exercise interventions to individual older patients who have been acutely hospitalized might reduce the pathophysiological disease process and restore physical functioning.

## Study strengths and limitations

The strengths of this study were the international panel with expertise in exercise interventions, the high response rate, the structured methodology and the relevance of the topic. The

study also had limitations. First, although the Delphi panel was chosen with care, all panel members were from Western countries, so recommendations from this study cannot be easily extrapolated to the healthcare systems of non-Western countries. Second, most panel members have a primary background in physical therapy, so the physical therapy profession may be overrepresented in the practice recommendations. This might have influenced the choice of the selected measurement tools or exercise intervention. However, the panel had a broad view on this topic and underscored the involvement of other healthcare professionals for optimal intervention.

## Conclusion

This Delphi study has provided starting points for developing an exercise intervention, COS and handover information that can prevent functional decline or restore physical functioning in older patients after discharge from hospital. The results of this Delphi study might help physical therapists to develop an exercise intervention for older patients with complex care needs after hospital discharge.

## Supporting information

**S1 Data.**
(DOCX)

**S1 File.**
(PDF)

## Acknowledgments

This study would not have been accomplished without the help of the panel of the 'Delphi consensus study on an exercise intervention for acutely hospitalized older patients', whose input in the three Delphi rounds made this study possible. The study panel consisted of the following members: A.W. Heinen, MSc, D. van Wijk, MSc, G. Pijnenburg, MSc, Prof. Dr. E.L. Cadore, Dr. C.J. Liu, Dr. C. McArthur, Prof. Dr. R. Daly, Dr. A.C. Bodilsen, Dr. N. Martinez-Velilla, Dr. E.V. Papa, Dr. J. Demarteau, Dr. M. Giné-Garriga, Dr. N. de Vries, Dr. M. Tieland, Prof. Dr. P. Calders.

## Author Contributions

**Conceptualization:** Jesse J. Aarden, Mel E. Major, Claartje M. W. Aghina, Martin van der Esch, Bianca M. Buurman, Raoul H. H. Engelbert, Marike van der Schaaf.

**Data curation:** Jesse J. Aarden, Martin van der Esch.

**Formal analysis:** Jesse J. Aarden.

**Funding acquisition:** Jesse J. Aarden.

**Investigation:** Jesse J. Aarden.

**Methodology:** Jesse J. Aarden, Mel E. Major, Claartje M. W. Aghina, Raoul H. H. Engelbert, Marike van der Schaaf.

**Project administration:** Jesse J. Aarden.

**Resources:** Jesse J. Aarden.

**Software:** Jesse J. Aarden.

**Supervision:** Jesse J. Aarden, Martin van der Esch, Bianca M. Buurman, Marike van der Schaaf.

**Validation:** Jesse J. Aarden, Marike van der Schaaf.

**Visualization:** Jesse J. Aarden, Mel E. Major.

**Writing – original draft:** Jesse J. Aarden, Claartje M. W. Aghina.

**Writing – review & editing:** Jesse J. Aarden, Mel E. Major, Martin van der Esch, Bianca M. Buurman, Raoul H. H. Engelbert, Marike van der Schaaf.

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
