## [Decision Letter · Decision Letter 0]

5 Jan 2023

PONE-D-22-27730Recommendations for an exercise intervention and core outcome set for older patients after hospital discharge: results of an international Delphi studyPLOS ONE

Dear Dr. Aarden,

Thank you for submitting your manuscript to PLOS ONE. After careful consideration, we feel that it has merit but does not fully meet PLOS ONE’s publication criteria as it currently stands. Therefore, we invite you to submit a revised version of the manuscript that addresses the points raised during the review process.

We look forward to receiving your revised manuscript.

Kind regards,

Jean-Philippe Regnaux, Ph.D, PT

Academic Editor

PLOS ONE

Journal Requirements:

Additional Editor Comments (if provided):

Dear Authors,

Your submission has been carefully read by reviewers. Their outputs are positifs. They have adressed minors comments which , i think , will improve the quality of your manuscript. Few minor changes or clarifications need to be done before it can be accepted for the publication. I hope to have the pleasure to reading you soon.

Reviewers' comments:

Reviewer's Responses to Questions

**Comments to the Author**

1. Is the manuscript technically sound, and do the data support the conclusions?

Reviewer #1: Yes

Reviewer #2: Yes

2. Has the statistical analysis been performed appropriately and rigorously? 

Reviewer #1: N/A

Reviewer #2: I Don't Know

3. Have the authors made all data underlying the findings in their manuscript fully available?

Reviewer #1: Yes

Reviewer #2: No

4. Is the manuscript presented in an intelligible fashion and written in standard English?

Reviewer #1: Yes

Reviewer #2: Yes

5. Review Comments to the Author

Reviewer #1: Thank you for the opportunity to review this really important Delphi study looking at exercise interventions and core outcomes for older people after hospital discharge.

My questions/responses are probably more thoughts about how the results of the Delphi might work in practice (particularly in a UK setting).

Abstract:

A clear overview of the study.

Introduction:

A good background and rationale for the study. I was just wondering though if you anticipate that the recommendations that you produce will be generalisable to healthcare systems across the world or was your focus on transitions from hospital to home in The Netherlands?

Methods:

Paragraph 1 - line 5: I am not clear what location AMC means or how it fits into this sentence?

You say there is a case description of an acutely hospitalized older adult and information from open questions in the supplementary material but I cannot find this information or am I missing something obvious?

Can you just clarify that by handover information you mean the information that is passed onto the healthcare professional/other who will be working with the older person in the community? Not information directly for the patient?

Results:

Clear and concise

Discussion:

Paragraph 2 - I totally agree that exercise and physical activity are not continued because of a lack of stimulus and self-discipline but there are also other factors including lack of community staff/support to follow-up these people (at least this is the case in the UK - hence my question about the scope of these results in countries other than the Netherlands) - so, by stimulus do you mean a person their to motivate the older person?

Conclusion:

I really appreciate that you have stated that this is a starting point and I really commend you on carrying out this work as a very important starting point

Reviewer #2: Manuscript review for: Recommendations for an exercise intervention and core outcome set for older patients after hospital discharge: results of an international Delphi study.

Many thanks for the opportunity to review the above-named manuscript. Whilst the aims and objectives for study were clear, I feel there are a few methodological and analytical points of note that may need to be addressed.

My comments are noted below.

Abstract

You added the keyword « resistance exercise training ». This term was only used in your abstract! It was not identified neither in the introduction nor in the results « exercise intervention » and figure 2. So, I am confused about the utility of this term. Please clarify

The authors mentioned in their abstract, line 6 “an internal panel of experts n=16” practically only 15 members responded so I think better to mention the number of responders instead.

Introduction

Further consideration is needed about this section and the order of the paragraphs and there needs to be a stronger rationale for the research.

Perhaps there needs to give some context to “acute hospitalization” (description and duration). Provide more context about exercise intervention (different types and programs) and its influence on functional decline with few examples and references. You mentioned that practical guidelines on FITT of home-based exercise intervention after hospitalization exists but are lacking! Citing the existing guidelines and giving more details about the content and its limitation could be helpful. Providing this sort would help to set the scene for the study more clearly and give significance to the results.

The authors mentioned in their introduction, line 9 “In this study, in-hospital exercise programs to prevent functional decline were performed twice per day”. Which study? The context is not clear! This part should be addressed.

Further information /details are needed to describe the need for this study.

Methods

A stronger narrative on the scoping literature review approach is required.

What were the search keywords used for the scoping literature review?

The search was limited on the last 10 years! Please justify this choice

Expert Panel. Selection criteria, homogeneity and panel size (minimum number of experts) were not mentioned in the “expert panel” part. More details are required

Panel members were identified as “experts”, how did you consider or measure experience? Based on years of clinical experience maybe! or number of publications in PubMed? (Cited in table 1) If yes, this should be included in the corresponding paragraph

The authors mentioned in the expert panel part, line 5 “Because no patients were included in this study, approval of the medical committee was not required.” Unsure if this is really useful, I prefer to delete this sentence.

Delphi rounds. How expert opinions were collected? Was it based on anonymous survey rounds, face-to-face or group encounters? Please clarify

How did you decide on the number and type of questions?

I would like to have a look on supplementary material on case description and additional questions used in this study.

Delphi round 3. The authors mentioned that “If an individual panel member’s scores differed from the panel’s median scores, they were asked to consider re-ranking the statement towards the median to reach consensus.” Not clear. Please explain more the procedure. Was it based on a panel discussion?

Results

Why did the authors contact only 16 experts and mainly physical therapists?

Theme 1. Daily exercise interventions, line 7 “1–2 times weekly interventions in the sub-acute phase” There is a disagreement with figure 2 where you mentioned that 2-3 times/week is required in sub-acute phase! Please clarify

The panel agreed on level intensity “70-80%” of the maximal heart rate. This requires elaboration about progressive training.

The authors presented the results regarding the consensus but what about the stability of responses between successive rounds? This information would be very interesting to explore

Figure 3 presents measurement tools across ICF domains. How do you explain the choice of these measurement tools knowing that other reliable scales exist in the literature?

Discussion

The discussion point about high-intensity exercise interventions. This requires elaboration.

I wonder if the novel finding of this study is “the expert consensus that exercise interventions should be tuned to the specific needs and goals of the patient.” Why was this information not identified in the results part?

Finally, its required to highlight what makes this research novel or unique.

6. PLOS authors have the option to publish the peer review history of their article (what does this mean?). If published, this will include your full peer review and any attached files.

Reviewer #1: No

Reviewer #2: No

---

## [Author Response · Author response to Decision Letter 0]

18 Feb 2023

Additional Editor Comments (if provided):

Response to PLOS ONE

Dear Authors,

Your submission has been carefully read by reviewers. Their outputs are positifs. They have adressed minors comments which , i think , will improve the quality of your manuscript. Few minor changes or clarifications need to be done before it can be accepted for the publication. I hope to have the pleasure to reading you soon.

Author response 

Thank you for the positive feedback and the possibility to improve the manuscript. We are grateful that we can response to the comments. The responses of the author are written in italics. 

Reviewers' comments to the author:

Reviewer #1: Thank you for the opportunity to review this really important Delphi study looking at exercise interventions and core outcomes for older people after hospital discharge.

My questions/responses are probably more thoughts about how the results of the Delphi might work in practice (particularly in a UK setting).

Abstract:

A clear overview of the study.

Introduction:

A good background and rationale for the study. I was just wondering though if you anticipate that the recommendations that you produce will be generalisable to healthcare systems across the world or was your focus on transitions from hospital to home in The Netherlands?

Author response

The starting point of the Delphi study was to include an international panel of experts. Thereforet the outcomes our generalizable to healthcare systems across the world. In the end, we included experts from seven different countries with five experts from the Netherlands. Primary reason that we included more experts from the Netherlands was that we wanted experts with practical expertise who still work in the field of geriatrics rehabilitation. In the end we think that the results will be generalizable although this should be adjusted to the different healthcare settings across the world.

Methods:

Paragraph 1 - line 5: I am not clear what location AMC means or how it fits into this sentence?

Author response

The Amsterdam UMC has 2 locations (AMC and VU) and we understand that this could be confusing. We deleted location AMC in the revised manuscript in the Methods section on page 6.

You say there is a case description of an acutely hospitalized older adult and information from open questions in the supplementary material but I cannot find this information or am I missing something obvious?

Author response

We included the case description in the supplementary material.

Can you just clarify that by handover information you mean the information that is passed onto the healthcare professional/other who will be working with the older person in the community? Not information directly for the patient?

Author response

Handover information is the information that is passed onto the healthcare professional from Hospital towards the primary care. We added ‘from hospital on to healthcare professionals in primary care.’ in the methods section on page 6.

Results:

Clear and concise

Discussion:

Paragraph 2 - I totally agree that exercise and physical activity are not continued because of a lack of stimulus and self-discipline but there are also other factors including lack of community staff/support to follow-up these people (at least this is the case in the UK - hence my question about the scope of these results in countries other than the Netherlands) - so, by stimulus do you mean a person their to motivate the older person?

Author response

We added the suggestion to the discussion on page 11: ‘After discharge from hospital, exercises and physical activity are often not continued because stimulus by staff or community21 and/or self-discipline17 are lacking.’

Conclusion:

I really appreciate that you have stated that this is a starting point and I really commend you on carrying out this work as a very important starting point

Author response

Thank you very much for all your feedback and we fully agree.

Reviewer #2: Manuscript review for: Recommendations for an exercise intervention and core outcome set for older patients after hospital discharge: results of an international Delphi study.

Manuscript review for: Recommendations for an exercise intervention and core outcome set for older patients after hospital discharge: results of an international Delphi study.

Many thanks for the opportunity to review the above-named manuscript. Whilst the aims and objectives for study were clear, I feel there are a few methodological and analytical points of note that may need to be addressed. 

My comments are noted below. 

Abstract

You added the keyword « resistance exercise training ». This term was only used in your abstract! It was not identified neither in the introduction nor in the results « exercise intervention » and figure 2. So, I am confused about the utility of this term. Please clarify

Author response

Thanks for the comment because the resistance exercise training is not correct. We changed in the abstract on page 4 resistance exercise training in exercise intervention as used in the rest of the manuscript: ‘The results of this Delphi study can assist physical therapists to provide a tailored exercise intervention for older patients with complex care needs after hospital discharge, to prevent functional decline and/or restore physical function.’

The authors mentioned in their abstract, line 6 “an internal panel of experts n=16” practically only 15 members responded so I think better to mention the number of responders instead.

Author response

We agree with the reviewer that describing the responders is clearer. We changed the abstract on page 4: ‘An internal panel of experts in the field of exercise interventions for acutely hospitalized older adults were invited to join the study.’ Later in the abstract it is described that there are fifteen responders.

Introduction 

Further consideration is needed about this section and the order of the paragraphs and there needs to be a stronger rationale for the research.

Perhaps there needs to give some context to “acute hospitalization” (description and duration).

Author response

In the first sentence of the introduction a short description and duration is added: For older adults, an acute hospitalization for multiple days due to an acute illness is a high-risk event with poor health outcomes, including functional decline, readmission, and mortality.1

Provide more context about exercise intervention (different types and programs) and its influence on functional decline with few examples and references. You mentioned that practical guidelines on FITT of home-based exercise intervention after hospitalization exists but are lacking! Citing the existing guidelines and giving more details about the content and its limitation could be helpful. Providing this sort would help to set the scene for the study more clearly and give significance to the results.

Author response

We added information to second paragraph of the introduction with reference: ‘International exercise recommendations in older adults are reported,18,19 These recommendations indicate that exercise improve physical function and quality of life and exercise is essential to older adults. However, practical guidelines on the frequency, intensity, time, and type (FITT) of home-based exercise interventions specifically for older patients after hospitalization are lacking.’

The authors mentioned in their introduction, line 9 “In this study, in-hospital exercise programs to prevent functional decline were performed twice per day”. Which study? The context is not clear! This part should be addressed. 

Author response

Accidently, the number of the reference was not included. We added reference 14 to the introduction: In this study14, in-hospital exercise programmes to prevent functional decline were performed twice per day.

Further information /details are needed to describe the need for this study.

Author response

At the end of the introduction on page 5, we added information to emphasize the novelty of this study: ‘In the absence of practical guidelines and high-quality randomized controlled trials focusing on acute hospitalized older adults, the Delphi methodology is often applied to obtain expert consensus on interventions for different populations.23,24’

Methods

A stronger narrative on the scoping literature review approach is required.

What were the search keywords used for the scoping literature review?

Author response

The scoping review was based on a search in Pubmed, Cinahl and Cochrane with MESH keywords such as aged, frail, hospitalization, hospital medicine, exercise therapy, resistance training, physical therapy modalities, diagnostic tests. Also, other keywords from key articles were used in the search. Snowballing was performed on relevant article to retrieve a complete overview of the literature.

The search was limited on the last 10 years! Please justify this choice

Author response

The choice for the last 10 years is arbitrary and was based on the changing healthcare in the last decade with shorter length of stay of patients after an acute hospitalization. 

Expert Panel. Selection criteria, homogeneity, and panel size (minimum number of experts) were not mentioned in the “expert panel” part. More details are required

Panel members were identified as “experts”, how did you consider or measure experience? Based on years of clinical experience maybe! or number of publications in PubMed? (Cited in table 1) If yes, this should be included in the corresponding paragraph

Author response

We wanted to select an international panel with a combination of expertise in this specific field of geriatric care. Eligibility of the expert panel was based on the field of expertise, geographical location, years of clinical expertise and the number of publications indexed in PubMed. 

The authors mentioned in the expert panel part, line 5 “Because no patients were included in this study, approval of the medical committee was not required.” Unsure if this is really useful, I prefer to delete this sentence.

Author response

Based on the feedback of the reviewer we removed this sentence.

Delphi rounds. How expert opinions were collected? Was it based on anonymous survey rounds, face-to-face or group encounters? Please clarify

Author response

Expert opinions were collected by a digital anonymous survey. We added to the Delphi rounds in the Methods section: ‘A digital survey was sent to generate ideas and to rank statements on a 9-point Likert scale, as per Delphi methodology recommendations.20‘

How did you decide on the number and type of questions? I would like to have a look on supplementary material on case description and additional questions used in this study.

Author response

The case description has been added to the supplementary material

Delphi round 3. The authors mentioned that “If an individual panel member’s scores differed from the panel’s median scores, they were asked to consider re-ranking the statement towards the median to reach consensus.” Not clear. Please explain more the procedure. Was it based on a panel discussion?

Author response

In the third round, individual panellists were asked if they were willing to adjust the score to the rest of the group. It was not possible to organize a physical meeting because of the international composition of the group 

Results

Why did the authors contact only 16 experts and mainly physical therapists?

Author response

The authors of this article wanted to focus on therapists who are working with older adults and have expertise in the field of exercise. Although most of the panel is (also) physical therapist, six out of fifteen of the included panel have another profession.

Theme 1. Daily exercise interventions, line 7 “1–2 times weekly interventions in the sub-acute phase” There is a disagreement with figure 2 where you mentioned that 2-3 times/week is required in sub-acute phase! Please clarify

Author response

The manuscript is adjusted in Theme 1 in the Results section: ‘Regarding training frequency, daily exercise interventions in the acute phase (up to 7 days post-discharge), 2–3 times weekly interventions in the sub-acute phase (up to 12 weeks post-discharge) and 1-12 times weekly in the long-term phase (>12 weeks post-discharge) were consensually ranked as essential for preventing functional decline.’

The panel agreed on level intensity “70-80%” of the maximal heart rate. This requires elaboration about progressive training.

Author response

It is interesting to elaborate more on the level of intensity of the maximal heart rate, however, this falls outside the scoop of the results of this Delphi study. In the discussion we added information to this finding (see author response in discussion).

The authors presented the results regarding the consensus but what about the stability of responses between successive rounds? This information would be very interesting to explore

Author response

The statements were ranked in the second round of the Delphi study and individual panellist were asked to reconsider their ranking to join the rest of the group and reach consensus. Therefore, for most of the panellist the responses did not change.

Figure 3 presents measurement tools across ICF domains. How do you explain the choice of these measurement tools knowing that other reliable scales exist in the literature?

Author response

The presented measurement tools are based on the input of the expert panel. Apparently, these measurement tools are frequently used in practice. It might be conceivable that other measurement tools are more used in research or academic setting. We added to the limitations of the discussion: ‘This might have influenced the choice of the selected measurement tools or exercise intervention.’

Discussion

The discussion point about high-intensity exercise interventions. This requires elaboration.

Author response

We agree that high-intensity training is an important discussion point. In the discussion we added: ‘Therefore, high intensity exercises can certainly be considered to restore physical function in line with the international exercise recommendation18’

I wonder if the novel finding of this study is “the expert consensus that exercise interventions should be tuned to the specific needs and goals of the patient.” Why was this information not identified in the results part? 

Author response

In the results section it is described as tailor-made exercise intervention. We added tailored in the discussion: ‘A novel finding of our study is the expert consensus that tailored exercise interventions should be tuned to the specific needs and goals (such as independent self-care, cooking or gardening) of the patient.’

Finally, its required to highlight what makes this research novel or unique.

Author response

In line with previous comments of the reviewers, we added information to the introduction to emphasize the novelty of this research. The novelty of this Delphi study is that this information was not available for older patients after discharge from hospital and can support healthcare providers in their daily practice.

---

## [Editor Report · Decision Letter 1]

12 Mar 2023

Recommendations for an exercise intervention and core outcome set for older patients after hospital discharge: results of an international Delphi study

PONE-D-22-27730R1

Dear Dr. Aarden,

We’re pleased to inform you that your manuscript has been judged scientifically suitable for publication and will be formally accepted for publication once it meets all outstanding technical requirements.

Kind regards,

Jean-Philippe Regnaux, Ph.D, PT

Academic Editor

PLOS ONE

Additional Editor Comments (optional):

I am pleased to inform you that your submission has been accepted for publication. Your study is very interesting. I am sure that the results will be useful to readers.
---

## [Editor Report · Acceptance letter]

15 Mar 2023

PONE-D-22-27730R1 

Recommendations for an exercise intervention and core outcome set for older patients after hospital discharge: results of an international Delphi study 

Dear Dr. Aarden:

I'm pleased to inform you that your manuscript has been deemed suitable for publication in PLOS ONE. Congratulations! Your manuscript is now with our production department. 

Kind regards, 

on behalf of

Mr Jean-Philippe Regnaux 

Academic Editor

PLOS ONE